# The Post-Traumatic Stress Disorder Checklist for DSM-5: Psychometric Properties of the Italian Version

**DOI:** 10.3390/ijerph19095282

**Published:** 2022-04-26

**Authors:** Marialaura Di Tella, Annunziata Romeo, Georgia Zara, Lorys Castelli, Michele Settanni

**Affiliations:** Department of Psychology, University of Turin, Via Verdi 10, 10124 Turin, Italy; marialaura.ditella@unito.it (M.D.T.); georgia.zara@unito.it (G.Z.); lorys.castelli@unito.it (L.C.); michele.settanni@unito.it (M.S.)

**Keywords:** post-traumatic stress symptoms, traumatic events, confirmatory factor analysis, Italian translation

## Abstract

Background: The present study aimed to investigate the psychometric properties of the Italian version of the PTSD Checklist for the DSM-V (PCL-5) in a group of adults who had experienced heterogenous traumatic events. Methods: Six hundred and one participants met the inclusion criteria and completed a set of questionnaires through an online survey. Before administering the survey, the PCL-5 was translated into Italian according to the back-translation method. The factorial structure of the PCL-5 was assessed through multiple confirmatory factor analyses. Gender measurement invariance and concurrent and criterion validity were also evaluated. Results: The instrument had a seven-factor structure and it worked in a similar manner for males and females. With regard to the concurrent validity, results showed that higher PCL-5 scores were associated with higher levels of depression and rumination and lower levels of life satisfaction. Regarding criterion validity, results revealed that PCL-5 scores were, on average, higher for females than for males, and the temporal distance from the traumatic event was negatively correlated with the total PCL-5 score. Conclusions: The findings indicated that the Italian version of the PCL-5 was able to provide valid and reliable scores for the assessment of PTSD symptoms in the Italian population.

## 1. Introduction

Post-traumatic stress disorder (PTSD) is a complex mental disorder resulting from exposure to extraordinarily traumatic events such as death or threatened death and actual or threatened serious injury or sexual violence [1].

PTSD is strongly associated with different psychological constructs and psychiatric disorders, and people with PTSD tend to experience lower levels of quality of life [2] and health satisfaction (Rauch et al., 2010). For example, the comorbidity between PTSD, depression and anxiety disorders has been widely documented in the specialized literature [3], and the role of rumination in both the development and maintenance of PTSD symptoms is well-established [4].

In the past few decades several assessment tools for trauma and related symptoms have been developed [5] due to increased interest in the identification and screening of PTSD. Although different screening tools are available for use in the assessment of PTSD, the PTSD Checklist (PCL) [6] is one of the most widely applied self-report measures in both clinical and research settings [7,8].

The diagnosis of PTSD has undergone major changes with the transition from the Diagnostic and Statistical Manual of Mental Disorders, Fourth Edition (DSM-IV) to the fifth edition of the Diagnostic and Statistical Manual of Mental Disorders (DSM-V). 

PTSD was moved from the anxiety disorders category to a new class of “trauma and stressor-related disorders,” and Criterion A2 (peri-traumatic fear, helplessness or horror) was eliminated. The new diagnosis includes 20 symptoms instead of the previous 17; three new symptoms were introduced (persistent and distorted blame of self or others, persistent negative emotional state and reckless or self-destructive behavior) and several others were revised [1].

The three symptom clusters in the DSM-IV (re-experiencing, avoidance and numbing and hyperarousal) were reorganized into four (intrusion, avoidance, negative alterations in cognition and mood and alterations in arousal and reactivity) by splitting the avoidance and numbing cluster into an avoidance cluster and a negative alteration in cognitions and mood cluster [1]. Subsequently, the PCL was recently updated in order to reflect revisions of the PTSD diagnostic criteria of the DSM-V. The new version, PCL-5 [6,9], includes 20 items that correspond to the 20 PTSD symptoms in the DSM-V. It is a self-report questionnaire that can be used to screen individuals for PTSD, monitor symptom changes and make a probable diagnosis of PTSD. Respondents are required to indicate how much they have been disturbed by symptoms during the previous month. The PCL-5 can be applied without considering criterion A (Exposure to actual or threatened death, serious injury or sexual violence in one (or more) of the following ways: 1. Directly experiencing the traumatic event(s); 2. Witnessing, in person, the event(s) as it occurred to others; 3. Learning that the traumatic event(s) occurred to a close family member or close friend. In cases of actual or threatened death of a family member or friend, the event(s) must have been violent or accidental; 4. Experiencing repeated or extreme exposure to aversive details of the traumatic event(s) (e.g., first responders collecting human remains, police officers repeatedly exposed to details of child abuse)) [1] and for each item, a score of 2 or above (range: 0 to 4) is considered clinically relevant. With regard to total symptom severity scores, preliminary findings are mixed, with values ranging from 28 to 38 [6,9].

Previous studies on the psychometric properties of the PCL-5 have analyzed different aspects of the reliability and validity of total PCL-5 scores. Total PCL-5 scores and subscale scores have shown high internal consistency, as well as convergent and divergent validity [9,10,11,12,13].

Specifically, results show high internal consistency both for the total scale (α = 0.90–0.97) and the four subscales (intrusions: α = 0.57–0.92; avoidance: α = 0.74–0.92; negative alterations in cognitions and mood: α = 0.78–0.92; hyperarousal: α = 0.75–0.90) [9,10,11,12,13]. The validity of the total PCL-5 score in terms of convergent and discriminant relationships with related or unrelated constructs has been evaluated in several studies. For example, the PCL-5 showed strong correlations with the measure of depression (*r* = 0.69–0.74) and more moderate correlations with anxiety (*r* = 0.40–0.72) [10,11,12,14,15]. Nevertheless, the PCL-5 score was found to be more strongly correlated with measures of related constructs (e.g., other measures of PTSD, depression) than those of unrelated constructs (e.g., personality features, substance abuse) [9,10,11,12,15]. Summing up, several findings showed good psychometric properties for the PCL-5.

Although previous studies provided evidence of the reliability and validity of the PCL-5, many study samples included limited populations (i.e., university students, military personnel, treatment-seeking civilians), and only two previous studies used a web-administered version of the PCL-5 [12,14].

The four-factor model of PTSD has been proposed by the DSM-V based on a series of confirmatory factor analytic (CFA) studies [8,16] that questioned the previous three-factor model. The new four-factor model has in turn been the subject of different CFA studies, which showed that the DSM-V model has poor fit when applied to PCL-5 data in most cases [17,18,19,20,21,22]. Even when an acceptable to good fit was demonstrated [11,15,18,21], there were other models that showed a significantly better fit.

Five alternative models to the three-factor DSM-IV model and the four-factor DSM-V model have been proposed over the past 10 years. Particularly, the Dysphoria Model and the Dysphoric Arousal Model were originally suggested by Simms et al. [23] and Elhai et al. [8], respectively, as alternative models to the three-factor DSM-IV model. Subsequently, two six-factor models (the Anhedonia Model [19]; the Externalizing Behavior Model [21]) and a seven-factor model (the Hybrid Model [17]) were proposed as different models for the four-factor DSM-V structure; for a detailed description of all of the factor structure models proposed in past studies, see Krüger-Gottschalk et al. [13] and Schmitt et al. [24].

The available evidence seems to demonstrate the superiority of the seven-factor Hybrid Model over both the original four-factor model [25] and the other alternative models that have been suggested [19,21]. Indeed, the seven-factor model has been found to have the best fit for the data in several previous studies [10,12,15,17,22].

Although the Hybrid Model appears to be the best fitting model, the literature on the latent structure of PTSD symptoms as evaluated by the PCL-5 is still unclear. For instance, Kruger-Gottschalk et al.’s [13] validation study on a sample with different traumatic experiences could not obtain interpretable results because of the linear dependence on the seven-factor model.

These contrasting results may be due to an intrinsic limitation of the seven-factor model. Indeed, four of the seven factors only have two items, which could result in various problems, such as inaccurate and unstable parameter estimates and difficulty in interpretation [7]. In order to overcome those possible limitations, Schmitt et al. [24] proposed a bifactor solution, which they found to be a more statistically appropriate model with respect to previously suggested PCL-5 factor structures. Another solution could be achieved through the use of a larger sample size of at least 400 participants, which can allow a fully valid and replicable solution to be obtained [17,26]. Finally, carrying out studies in different geographic areas might help define the structure of PTSD symptoms while considering the potential effects of cross-cultural variations.

The main aim of the present study was to examine the psychometric properties of the Italian version of the PCL-5. In order to reach this goal, we first tested which among the main factor models that have been proposed for the PCL-5 was the best fit for our data. Secondly, gender measurement invariance was assessed. Findings from previous studies revealed that gender differences exist in PTSD [27]. However, to ascertain whether between-group differences really exist, each sample must have a similar understanding of the questionnaire’s items; that is, measurement invariance across gender needed to be established. Next, we evaluated the concurrent validity of the PCL-5 with respect to intrusive and deliberate rumination, depressive symptoms and life satisfaction, in accordance with the available evidence that seems to show significant associations between those constructs and PTSD symptoms [2,15,28,29,30]. Finally, we further assessed the criterion validity of the instrument by testing for the presence of gender differences in the scores obtained and by examining the relationship between PCL-5 scores and the time since the traumatic event.

## 2. Materials and Methods

### 2.1. Participants and Procedure

In order to qualify for the study, participants had to meet the following inclusion criteria: being at least 18 years of age; being a native Italian speaker; having a sufficient educational level (>5 years); having experienced at least one traumatic event in the past 10 years based on the DSM-V criteria (see criterion A definition above).

Seven hundred and fifty-nine participants were contacted, 688 completed the survey and 601 met the inclusion criteria (response rate: 79%) and constituted the final sample of the study. Participants who did not meet the inclusion criteria were excluded from the survey system in the initial stages.

The present data were collected using an anonymous online survey from 13 March 2018 to 16 August 2019. A snowball sampling strategy was employed, wherein the participants were initially made aware of the research via online advertisements. Participants interested in the study emailed the researchers who provided them with the survey link, and participants were encouraged to disseminate the link to others. Participants were recruited from the general Italian population (from different regions of Italy). Before administering the questionnaires, the PCL-5 and the Event-Related Rumination Inventory were translated into Italian according to the back-translation method to ensure the semantic equivalence of the Italian and English versions. Accordingly, the two measures were initially translated from English into Italian by two experts in the field with fluent English, and back-translated by an English university lecturer with fluent Italian. The two English versions for each measure were finally compared and differences were identified and corrected.

Afterwards, an anonymized, individual and unique code was emailed to those who agreed to take part in the study (by providing written informed consent) to complete the online survey. Before completing the questionnaires, all participants were asked to provide sociodemographic (i.e., age, gender, educational level and marital status) and trauma-related information (i.e., definition of stressful event as trauma-inclusion criterion; type of traumatic event; time since the traumatic event). In particular, a list of possible traumatic experiences was included in the survey and participants were asked to choose one of them.

The study was approved by the University of Turin Ethics Committee (Prot. n. 264810) and was conducted in accordance with the Declaration of Helsinki.

### 2.2. Measures


*Post-traumatic stress symptoms*


The PCL-5 [9,31] is a 20-item self-report measure that has been developed for the assessment of PTSD symptoms. Each item is scored using a 5-point Likert-type scale ranging from 0 (not at all) to 4 (extremely). Respondents are required to indicate the degree to which they have been bothered in the past month by DSM-V PTSD symptoms related to their most currently distressing event [31]. The total score ranges from 0 to 80, with higher scores indicating higher levels of PTSD symptoms.


*Intrusive and Deliberate Rumination*


Intrusive and deliberate rumination in relation to the traumatic event identified by the participants was assessed using the Italian translation of the Event-Related Rumination Inventory (ERRI) [32]. ERRI is a 20-item self-report measure, which reflects two kinds of rumination: recent intrusive rumination (e.g., “In the past two weeks, thoughts about the event came to mind and I could not stop thinking about them”) and recent deliberate rumination (e.g., “In the past two weeks, I thought about the event and tried to understand what happened”). Each item is scored using a 4-point Likert-type scale ranging from 0 (not at all) to 3 (often). Two separate total scores can be derived for intrusive and deliberate rumination (i.e., ERRI-intrusive rumination and ERRI-deliberate rumination). Each total score ranges from 0 to 30, with higher scores indicating more intrusive or deliberate rumination.

The scale has shown good internal consistency, with Cronbach’s alpha coefficients ranging from 0.88 to 0.96 [32,33,34]. In line with these results, in our sample, the Cronbach’s alpha values were good for both the ERRI-intrusive rumination (α = 0.97) and the ERRI-deliberate rumination (α = 0.95).


*Depressive symptoms*


The presence of depressive symptoms was assessed using the Beck Depression Inventory-II (BDI-II) [35,36]. It consists of 21 items, each scored using a 4-point Likert-type scale ranging from 0 (no symptoms) to 3 (most severe). The total score is the sum of all the items and ranges from 0 (no depressive symptoms) to 63 (severe depression).

The BDI-II has shown good psychometric properties, with good internal consistency (Cronbach’s α score = 0.91), test-retest reliability and construct validity [37]. In line with these results, in our sample, the Cronbach’s alpha was excellent for the BDI-II (α = 0.93).


*Satisfaction with life*


For an index of life satisfaction, a Visual Analogue Scale (VAS) ranging from 0 (Not satisfied at all) to 10 (Completely satisfied) was used to assess the average life satisfaction of participants. Single-item life satisfaction measures have demonstrated substantial degree of criterion validity with largely employed scales such as the Satisfaction with Life Scale [38].

### 2.3. Statistical Analyses

Validation analyses of the instrument in the Italian context were conducted as follows. First, we examined the frequencies of individual items to identify any unexpected functioning of individual items and assessed factorability using the Bartlett’s test of sphericity and the Kaiser−Myer−Olkin (KMO) statistic. Next, we conducted confirmatory factor analyses to determine which of the different factorial structures proposed in the literature best fit the data collected in the Italian context. In particular, five alternative measurement models were examined: 1-dimensional, 4 related factors, 7 related factors, second order and bifactor models. For the estimation of the parameters, the MLR estimator was used. We did not use the WLSMV estimator because the use of MLR allows one to calculate additional fit statistics (AIC and BIC) that in turn allow one to compare the fit of non-nested models. For further control, the analyses were also performed with the WLSMV estimator which led to similar results. The fit of the models was evaluated using the Approximate Root Medium Square Error (RMSEA), Comparative Fit Index (CFI), Tucker−Lewis Index (TLI) and Standard Root Mean Square Residual (SRMR).

The following criteria for evaluating model fit were used: RMSEA ≤ 0.06, CFI ≥ 0.95, TLI ≥ 0.95 and SRMR < 0.08 for good fit and RMSEA ≤ 0.10, CFI ≥ 0.90, TLI ≥ 0.90 and SRMR < 0.10 for adequate fit [39].

To compare the fit of different tested models, the scalar−to−scalar difference test (Δχ2), AIC and BIC were used to compare the nested models, while comparisons with the bifactor model were performed using only AIC and BIC. Value differences of more than 10 on AIC and BIC indicate a relevant difference in model adaptation [40,41], with lower values indicating a better fit.

Next, measurement invariance by gender was tested. This was pursued by sequentially testing the different types of invariance (configural, metric and scalar) by conducting successive multigroup CFAs. This permits testing stability of scores to support the invariance by setting cross-group constraints and comparing more-restricted with less-restricted models. To identify significant differences between models, we followed Chen’s recommendations [42]: a change in the CFI and the RMSEA equal to or greater than 0.010 would be indicative of non-invariant scores.

The reliability of the instrument was then calculated using Cronbach’s alpha index of internal consistency. Reliability indices were computed both for the total scale score and for its subscales.

As a further test of concurrent validity, correlations between the total PCL-5 score and depression, life satisfaction and rumination scores (measured concurrently) were calculated. Finally, the criterion validity of the instrument was examined by testing for gender differences and by correlating the obtained PCL-5 score with the variable, “months since the traumatic event”. All analyses were conducted using SPSS 27.0 and Mplus 8.0 [43].

## 3. Results


**
*Sociodemographic and trauma-related data*
**


Sociodemographic data from the Italian sample are presented in Table 1. The majority of participants were women (429; 71.4%), and the mean age of the sample was 30.9 years (SD = 11.6). The age range of the participants was 18–72 years.

Participants also reported that 39.8 (± 32.8) months had passed since the traumatic event. The majority of participants (226; 37.6%) experienced the death of a relative or friend.


**
*Item functioning*
**


The mean of the total PCL-5 score was 23.5 (SD = 18.4). Descriptive analyses of the responses to individual PCL-5 items and the total score are shown in Table 2.


**
*Factor structure and measurement invariance*
**


Before carrying out the factor analyses, we tested the factorability of the PCL items. The KMO measure of sampling adequacy (KMO = 0.957) supports optimal common variance for factor analysis. In addition, Bartlett’s test for sphericity was significant (χ^2^ (190) = 7835.39; *p* < 0.01), indicating that the correlation matrix is significantly different from the identity matrix and therefore, suitable for structure detection (see Appendix A for the correlation matrix among the PCL-5 items).

In order to examine the factor structure of the Italian PCL-5, five different models suggested in the literature were tested. In particular, we examined the fit of the following factor structure models (see Table 3 for a detailed item mapping of the tested models): 1. Unidimensional model; 2. DSM-V 4-factor model; 3. 7-factor Hybrid Model; 4. Second order model (a hierarchical model with a second-order general PTS factor influencing the 7 specific PTS latent factors that in turn influence the item responses); 5. Bifactor model (a model in which both a general PTS factor and 7 specific factors exist and they additively influence the item responses).

Fit statistics for the tested models are reported in Table 4. The CFA analysis for the unidimensional model showed an inadequate fit, while the fit for the DSM-V (4-factor) model was adequate. However, the other tested models (i.e., 7-factor, second-order and bifactor models) showed a better fit. In particular, the 7-factor model achieved a significantly better fit than the other models, as tested by a chi-square difference test for comparison with nested models and ΔBIC for the bifactor model (ΔAIC = 69.18; ΔBIC = 33.98). Factor loadings for the best-fitting (7-factor) model are shown in Appendix A (see Appendix A).

Measurement invariance across genders was tested by conducting successive multi-group CFA models. Statistics reported in Table 5 indicate sufficient support to assume configural, metric and scalar invariance.


**
*Reliability*
**


Cronbach’s alpha coefficient of the total score was 0.95. The Cronbach’s coefficient of subscale scores computed according to the 7-factor model was 0.88 for re-experiencing, 0.77 for avoidance, 0.81 for negative affect, 0.86 for anhedonia, 0.63 for externalizing behavior, 0.82 for anxious arousal and 0.75 for dysphoric arousal.


**
*Concurrent and criterion validity*
**


The concurrent validity of the instrument was analyzed by calculating the value of the Pearson’s correlations between the total score of the PCL-5 and the scores of depression (BDI-II), life satisfaction and deliberate and non-deliberate rumination. The correlations were equal to 0.67 (with BDI score), −0.46 (with life satisfaction), 0.54 (with deliberate rumination score) and 0.62 (with intrusive rumination score).

Criterion validity was also tested by comparing male and female total PCL-5 scores: we found that females scored higher than males (Males: M = 18.95, SD = 16.58; females: M = 25.28, SD = 16.58; *t*(599) = 3.87, *p* < 0.001, ES: Cohen’s *d* = 0.35). Lastly, we computed the correlation between total PCL-5 score and ‘months since the traumatic event’. The correlation coefficient was *r* = −0.19. The negative correlation indicates that the more recent the traumatic event, the greater the expected degree of the PCL-5 score.

## 4. Discussion

The present study aimed to examine the psychometric properties of the Italian version of the PCL-5 in a heterogeneous sample of Italian adults who experienced a traumatic event during the last 10 years. Specifically, the best factor structure for the Italian translation, gender measurement invariance and concurrent and criterion validity were assessed.

Particularly, we found that the instrument, when applied to the Italian context, proved to have a seven-factor structure. In fact, although the original DSM-V model showed an adequate fit, the Hybrid Model was found to have a superior fit, consistent with recent validation studies conducted in different countries and populations (e.g., military personnel, University students and treatment-seeking cohorts) [9,10,12,15,17,22,44].

Despite the available evidence in its favor, the seven-factor Hybrid Model has been questioned by some researchers as the inclusion of so many factors might be statistically problematic. Indeed, four of the seven factors only have two items, and this can make the composite scores for these factors unreliable [45] and the parameter estimation unstable, particularly in small sample sizes [17]. However, it must be noted that both the original DSM-V four-factor model and the other suggested models of PTSD are comprised of factors with two items (e.g., the avoidance factor). Furthermore, the superior fit of the seven-factor model over the other proposed models has been demonstrated in a growing body of research. The present results extend the available evidence to a large sample of Italian participants who reported various traumatic experiences, increasing the credibility of the Hybrid Model as a possible, alternative conceptualization of PTSD symptomatology. However, additional evidence is necessary to support these findings and to establish the predictive validity and clinical utility of this more extensive model of PTSD.

The instrument also shows it works comparably for both males and females, having been verified by the presence of configural, metric and scalar measurement invariance.

In addition, the instrument showed adequate reliability for all of the PCL-5 subscales, with the only exception being the “externalizing behavior” subscale, which reported a lower Cronbach’s alpha value compared to the others.

Furthermore, the validity of the instrument is supported by the pattern of correlations found between the total score and theoretically-related constructs. Particularly, concurrent validity was tested by correlating the total PCL-5 score with BDI-II, ERRI and life satisfaction scores. Results have shown that higher PCL-5 scores are associated with higher levels of depression and rumination (both deliberate and non-deliberate) and lower levels of life satisfaction. The positive correlation between depression and PTSD has been widely reported in previous studies [8,14,29,46]. Indeed, not only are there several overlapping symptoms between depression and PTSD, but these two psychopathological conditions are also frequently comorbid [47]. Similarly, it is not surprising to see the strong association between rumination and PTSD symptoms, which has already been highlighted by previous systematic reviews [4,20]. In fact, researchers have suggested that trauma-related rumination is a form of cognitive avoidance that may lead to the development and maintenance of PTSD [48]. Finally, in line with the available literature [2,30], our results showed that people with higher levels of PCL-5 experienced lower levels of life satisfaction, suggesting that PTSD and life satisfaction could be considered two separate and opposite constructs.

As far as criterion validity is concerned, differences between men and women on the total PCL-5 score, and correlations between the total PCL-5 score and the months since the traumatic event were computed. Results showed that PCL-5 scores were, on average, higher for females than for males, and temporal distance from the traumatic event was negatively correlated with the total PCL-5 score. These results are consistent with findings from several previous studies [49,50,51] which displayed the higher prevalence of PTSD symptoms in women than men. Particularly, the prevalence in women has been found to be approximately twice as high than in men and several factors can contribute to sex differences in PTSD [27]. The relationship between the time since the traumatic event and PTSD is much more complex and mediated by several psychological and trauma-related factors [52,53,54].

The present study is not exempt from some limitations that should be acknowledged. First, a large proportion of the sample was comprised of female and relatively young participants. Second, although both the present and previous studies seem to support the concurrent validity of the PCL-5 [9,10,11,12,14,15], we did not employ other instruments of PTSD symptoms to establish concurrent validity of the questionnaire more thoroughly. Moreover, the absence of a structured interview for the evaluation of PTSD did not allow us to examine the interpretability of the Italian version of the PCL-5. Finally, we were not able to include a second time-point assessment for our sample of Italian participants in order to evaluate test-retest reliability. Further studies recruiting a greater number of male and older participants and examining the reliability and validity of the Italian version of the PCL-5 more deeply should be carried out. This would support the generalizability of our results and the psychometric properties of the instrument.

## 5. Conclusions

Despite these limitations, this is the first study validating the Italian version of the PCL-5. The current findings suggest that the Italian version of the PCL-5 was able to provide valid and reliable scores for the assessment of PTSD symptoms in an adult, Italian population who experienced different traumatic events.

## Figures and Tables

**Table 1 ijerph-19-05282-t001:** Sociodemographic and trauma-related data of the Italian participants (*N* = 601).

	Mean (SD)	*n* (%)	Range
Age (years)	30.91 (11.72)		18–72
**Gender**			
Female		429 (71.4)	
Male		172 (28.6)	
**Education**			
Basic education (ISCED 1/2)		20 (3.4)	
Secondary education (ISCED 3/4)		210 (34.9)	
Tertiary education (ISCED 5/6)		371 (61.8)	
**Marital status**			
Never married		432 (71.9)	
Cohabitant/Married		140 (23.3)	
Separated/Divorced		21 (3.5)	
Widowed		8 (1.3)	
**Types of traumatic events**			
Being involved in a serious accident		42 (7.0)	
Being stalked		23 (3.8)	
Death of a relative/friend		226 (37.6)	
Natural disaster		63 (10.5)	
Others		72 (12.1)	
Physical or sexual assault		30 (5.0)	
Robbery or mugging		10 (1.7)	
Serious illness of a relative/friend		73 (12.2)	
Serious medical condition		62 (10.3)	

SD = Standard Deviation.

**Table 2 ijerph-19-05282-t002:** PCL-5 descriptive statistics.

PCL-5 Item	Item Description	M	SD	Skewness	Kurtosis	Range
1	Memories	1.32	1.17	0.67	−0.38	0–4
2	Dreams	0.91	1.15	1.18	0.42	0–4
3	Flashbacks	0.91	1.15	1.18	0.50	0–4
4	Cued distress	1.58	1.31	0.35	−0.97	0–4
5	Cued physical reactions	1.15	1.29	0.86	−0.45	0–4
6	Avoiding internal reminders	1.34	1.26	0.58	−0.72	0–4
7	Avoiding external reminders	1.27	1.35	0.73	−0.72	0–4
8	Dissociative amnesia	0.91	1.14	1.17	0.48	0–4
9	Negative beliefs	1.04	1.32	1.07	−0.08	0–4
10	Blame	1.12	1.39	0.94	−0.51	0–4
11	Negative feelings	1.47	1.37	0.56	−0.92	0–4
12	Loss of interest	1.03	1.28	1.00	−0.18	0–4
13	Detachment or estrangement	1.12	1.33	0.90	−0.44	0–4
14	Numbing	1.09	1.28	0.93	−0.33	0–4
15	Irritability or aggressive behavior	1.27	1.30	0.79	−0.48	0–4
16	Reckless behavior	0.67	1.00	1.73	2.33	0–4
17	Hypervigilance	1.48	1.30	0.51	−0.77	0–4
18	Startle	1.27	1.31	0.72	−0.64	0–4
19	Concentration	1.29	1.34	0.73	−0.67	0–4
20	Sleep	1.22	1.35	0.83	−0.56	0–4
	Total score	23.47	18.35	0.73	−0.20	0–80

**Table 3 ijerph-19-05282-t003:** Item mapping of the main alternative factorial models proposed for the PCL-5.

Items	DSM-V Model (4-Factor)	DSM-IV Dysphoria Model (4 Factors) *	DSM-IV Dysphoric Arousal Model (5 Factors) *	Anhedonia Model (6 Factors)	Externalizing Behavior Model (6 Factors)	Hybrid Model (7 Factors)	1-Factor Model
1. Intrusive thoughts	R	R	R	R	R	R	P
2. Nightmares	R	R	R	R	R	R	P
3. Flashbacks	R	R	R	R	R	R	P
4. Emotional cue reactivity	R	R	R	R	R	R	P
5. Physical cue reactivity	R	R	R	R	R	R	P
6. Avoidance of thoughts	AV	AV	AV	AV	AV	AV	P
7. Avoidance of reminders	AV	AV	AV	AV	AV	AV	P
8. Trauma-related amnesia	NACM	D	NACM	NACM	NACM	NA	P
9. Negative beliefs	NACM	D	NACM	NACM	NACM	NA	P
10. Distorted blame	NACM	D	NACM	NACM	NACM	NA	P
11. Persistent negative emotional state	NACM	D	NACM	NACM	NACM	NA	P
12. Lack of interest	NACM	D	NACM	AN	NACM	AN	P
13. Feeling detached	NACM	D	NACM	AN	NACM	AN	P
14. Inability to experience positive emotions	NACM	D	NACM	AN	NACM	AN	P
15. Irritability or anger	AR	D	DA	DA	EB	EB	P
16. Recklessness	AR	AR	DA	DA	EB	EB	P
17. Hypervigilance	AR	AR	AA	AA	AA	AA	P
18. Exaggerated state	AR	AR	AA	AA	AA	AA	P
19. Difficulty concentrating	AR	D	DA	DA	DA	DA	P
20. Sleep disturbance	AR	D	DA	DA	DA	DA	P

AA = anxious arousal: AN = anhedonia; AR = alterations in arousal and reactivity; AV = avoidance; D = dysphoria; DA = dysphoric arousal; EB = externalizing behaviors; NA = negative affect; NACM = negative alterations in cognitions and mood; P = general PTSD factor; R = re-experiencing. ***** Modified versions of the original DSM-IV Dysphoria and Dysphoric Arousal models based on the different and additional symptoms of DSM-5.

**Table 4 ijerph-19-05282-t004:** Model fit for tested factor structure models.

Model	Chi-Square	Scaling Correction	df	RMSEA	[90% CI]	CFI	TLI	SRMR	AIC	BIC
Unidimensional	935.618	14.168	170	0.087	0.081–0.092	0.86	0.84	0.05	33,294.42	33,558.34
DSM-V (4-factor)	569.719	1.411	164	0.064	0.058–0.070	0.92	0.91	0.04	32,784.56	33,074.86
7-factor	386.87	1.398	149	0.052	0.045–0.058	0.96	0.94	0.04	32,551.67	32,907.96
Second-order	476.405	1.4	163	0.057	0.051–0.062	0.94	0.93	0.04	32,650.11	32,944.81
Bifactor	451.52	1.39	157	0.056	0.050–0.062	0.95	0.93	0.04	32,620.85	32,941.94

**Table 5 ijerph-19-05282-t005:** Measurement Invariance: Model fit indices and model comparisons. BELOW figures out of line in item 1.

	Model Fit				Model Comparisons	
Model Tested	χ^2^ (df)	Δχ^2^ (df)	RMSEA	CFI	ΔRMSEA	ΔCFI
Configural Invariance	562.375 (298)		0.054	0.95	0	0
Metric Invariance	579.17 (311)	−13	0.054	0.95	0	0
Scalar invariance	602.495 (324)	−13	0.053	0.948	0.001	0.002

## Data Availability

Not applicable.

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
