# Peer review of "The Post-Traumatic Stress Disorder Checklist for DSM-5: Psychometric Properties of the Italian Version"

_ijerph, 2022, doi:10.3390/ijerph19095282_

Round 1

Reviewer 1 Report

The presented article entitled ‘The Posttraumatic Stress Disorder Checklist for DSM-5: Psychometric properties of the Italian version’ is a valuable study presenting psychometric research on the Italian version of the tool that can be helpful in conducting research on PTSD symptoms in the Italian population, which is especially valuable in times of the COVID-19 post-pandemic period. The article is well thought out, presented in a logical concept, with a good selection of methods for assessing the preliminary psychometric properties of the tool.

In the article, it would be worth correcting a few shortcomings before publication:

- the article should be checked again in terms of language. The article contains sentences with an incorrect grammatical or stylistic structure, e.g. lines 41-42, 71-73, 200-201.

- in line 67 the authors refer to criterion A, which, however, they do not explain.

- After the sentence in line 73, a citation should be indicated.

- In material and methods section, there is no information about the period in which the research was conducted.

- What events were considered traumatic? The article does not discuss the criteria for inclusion in the study group. What were the criteria for recognizing that the subjects had experienced a traumatic experience?

- The article describes the procedure of gathering the study group too poorly. Was the survey addressed to a specific group, or were emails with requests to participate sent randomly? If so, how was the database of candidates for the study prepared?

- There is relatively little information about the study group. Due to the lack of a sufficient description, questions arise whether people from the general population were examined? Are they from one region? Etc.

- Sometimes there is a lack of consistency in the data, e.g. lines 196-197 (Cronbach's α score = 0.91; α = .93). In cases where the value is below 1, I recommend that you use .xx instead of 0.xx.

- The presentation of table 5 should be considered. Currently, it covers almost the entire page, with only a few numerical information.

Thank you for the opportunity to review this interesting article. I hope that these comments will help to improve the quality of the manuscript.

Author Response

Reviewer 1

General comment
The presented article entitled ‘The Posttraumatic Stress Disorder Checklist for DSM-5: Psychometric properties of the Italian version’ is a valuable study presenting psychometric research on the Italian version of the tool that can be helpful in conducting research on PTSD symptoms in the Italian population, which is especially valuable in times of the COVID-19 post-pandemic period. The article is well thought out, presented in a logical concept, with a good selection of methods for assessing the preliminary psychometric properties of the tool.

Thank you for your appreciation and for your insightful comments. We have tried to take into account all your suggestions in order to improve the quality of our manuscript.

Specific comments:

In the article, it would be worth correcting a few shortcomings before publication:

- the article should be checked again in terms of language. The article contains sentences with an incorrect grammatical or stylistic structure, e.g. lines 41-42, 71-73, 200-201.

We are grateful for this suggestion. We have now improved the quality of our manuscript, revising the English language.

- In line 67 the authors refer to criterion A, which, however, they do not explain.

We have now added a definition of criterion A in line 67.

- After the sentence in line 73, a citation should be indicated.

We thank you for this suggestion, and we have now included references for the sentence in line 73.

- In material and methods section, there is no information about the period in which the research was conducted.

We have now included the recruitment period in the “Materials and Methods” section.

- What events were considered traumatic? The article does not discuss the criteria for inclusion in the study group. What were the criteria for recognizing that the subjects had experienced a traumatic experience?

We agree with your comment. We have in fact discussed the inclusion criteria, with particular regard to the traumatic events the participants experienced.

- The article describes the procedure of gathering the study group too poorly. Was the survey addressed to a specific group, or were emails with requests to participate sent randomly? If so, how was the database of candidates for the study prepared?

Thank you for this suggestion and we apologise for not having been clear on this aspect. We have now specified the participants’ recruitment. In the manuscript we have explained that (1) participants who agreed to take part in the study were emailed the survey link; (2) the survey was set up to ask participants a series of preliminary questions to ascertain their eligibility for the study. Participants who did not meet the inclusion criteria were automatically excluded by the system.

- There is relatively little information about the study group. Due to the lack of a sufficient description, questions arise whether people from the general population were examined? Are they from one region? Etc.

Thank you for this comment. In the revised version of our manuscript the participants’ recruitment process is now presented clearly and more completely. It has been specified that participants were recruited from the general Italian population.

- Sometimes there is a lack of consistency in the data, e.g. lines 196-197 (Cronbach's α score = 0.91; α = .93). In cases where the value is below 1, I recommend that you use .xx instead of 0.xx.

We apologise for the lack of consistency in data presentation. We have now reported homogeneously Cronbach’s alpha values.

- The presentation of table 5 should be considered. Currently, it covers almost the entire page, with only a few numerical information.

Thank you for your suggestion. We have now included Table 5 as Supplementary Material.

Thank you for the opportunity to review this interesting article. I hope that these comments will help to improve the quality of the manuscript.

We thank you once more for your constructive comments and suggestions.

Reviewer 2 Report

The manuscript entitled "The Posttraumatic Stress Disorder Checklist for DSM-5: Psychometric properties of the Italian version" is an important adaptation and validation study examining PCL-5 in Italian culture. The manuscript shows that a hybrid 7-factor model is the most appropriate for the sample. Introduction is concise, but comprehensive. Unfortunately, the Methods section is poorly described and does not allow replication of the study. Also, results can be improved. Some suggestions are presented below:

  1. It is unclear how participants were recruited. Authors stated (lines 159-160): "Afterwards, an anonymized, individual and unique code to complete the online survey was emailed to those who gave their agreement to take part in the study". How did you find email addresses to send invitations to research? How many emails did you send? How many people refused to participate in the study? How many people did not meet inclusion criteria? Could you list criteria and give the frequencies of excluded people for each criterion? What was a response rate in the survey?
  2. Indeed, the most unclear is the inclusion criteria. What stage of the study was to exclude people who did not meet criteria? How did you control each criterion, in particular, for having experienced at least one traumatic event in the past 10 years based on DSM-5 criteria? Could you list all traumatic events you found, and the frequency of each in your participants sample? How did you ask about it? Did you show a list of events was presented, or people answered an open question?
  3. It is unclear what "basic, secondary, and tertiary education" means in Italy. Please add more information explaining the categories of education in Table 1.
  4. Could you please give the statistics for the Bartlett's test of Sphericity and KMO? Could you show the correlation matrix between all questions in the PLC-5 (e.g., a heatmap would be helpful).
  5. Could you comment in the Discussion section a low reliability (α = 0.63) for externalizing behavior?

Minor revision: Chen (2007) should be listed in References (line 232).

Author Response

General comment
The manuscript entitled "The Posttraumatic Stress Disorder Checklist for DSM-5: Psychometric properties of the Italian version" is an important adaptation and validation study examining PCL-5 in Italian culture. The manuscript shows that a hybrid 7-factor model is the most appropriate for the sample. Introduction is concise, but comprehensive. Unfortunately, the Methods section is poorly described and does not allow replication of the study.

Thank you for your appreciation and for your insightful comments. We have taken into account all your constructive suggestions in order to improve the quality of our manuscript.

Specific comments:

Also, results can be improved. Some suggestions are presented below:

  1. It is unclear how participants were recruited. Authors stated (lines 159-160): "Afterwards, an anonymized, individual and unique code to complete the online survey was emailed to those who gave their agreement to take part in the study". How did you find email addresses to send invitations to research? How many emails did you send? How many people refused to participate in the study? How many people did not meet inclusion criteria? Could you list criteria and give the frequencies of excluded people for each criterion? What was a response rate in the survey?

As critically suggested, we have now clarified the participants’ recruitment process and indicated the number of participants who completed the survey and met the inclusion criteria. We have also computed the response rate for the survey and included it in the revised version of our manuscript.

  1. Indeed, the most unclear is the inclusion criteria. What stage of the study was to exclude people who did not meet criteria? How did you control each criterion, in particular, for having experienced at least one traumatic event in the past 10 years based on DSM-5 criteria? Could you list all traumatic events you found, and the frequency of each in your participants sample? How did you ask about it? Did you show a list of events was presented, or people answered an open question?

We have now clarified the participants’ recruitment process in the revised version of our manuscript. We have also indicated the list of traumatic events that we included in the survey and the frequency for each of them.

  1. It is unclear what "basic, secondary, and tertiary education" means in Italy. Please add more information explaining the categories of education in Table 1.

We used the ISCED (International Standard Classification of Education) system to classify the educational level of the participants involved in the study. Primary education corresponds to Italian elementary and middle school education; secondary education corresponds to higher education; tertiary education corresponds to Bachelor's and Master's degrees and Postgraduate degrees.

  1. Could you please give the statistics for the Bartlett's test of Sphericity and KMO? Could you show the correlation matrix between all questions in the PLC-5 (e.g., a heatmap would be helpful).

We have now included the statistics for the Bartlett's test of Sphericity and KMO. We have also added a table (Appendix B) with the correlation matrix between all questions in the PLC-5 in the revised version of our manuscript (please see Supplementary Material).

  1. Could you comment in the Discussion section a low reliability (α = 0.63) for externalizing behavior?

Thank you for your suggestion. We have now included a brief comment about the lower reliability value we found for the externalising behaviour compared to the other PCL-5 subscales.

  1. Minor revision: Chen (2007) should be listed in References (line 232).

Thank you for this observation. We have included this missing citation in the reference list.

Round 2

Reviewer 2 Report

The authors have corrected the manuscript and it is ready for publication now.